Identification of areas of very high biodiversity value to achieve the EU Biodiversity Strategy for 2030 key commitments

Miu Iulia V. 1
Rozylowicz Laurentiu laurentiu.rozylowicz@g.unibuc.ro 1 2
Popescu Viorel D. 1 3
Anastasiu Paulina 4
1 Center for Environmental Research, University of Bucharest , Bucharest , Romania
2 Chelonia Romania , Bucharest , Romania
3 Department of Biological Sciences, Ohio University , Athens , OH , United States of America
4 Dimitrie Brândză Botanical Garden, University of Bucharest , Bucharest , Romania
Colla Sheila
Electronic publication date: 2020 Sep 30
Publication date: 2020
Volume: 8
Electronic Location ID: e10067
Received 2020 Jul 14; Accepted 2020 Sep 8
Copyright: ©2020 Miu et al.
Copyright year: 2020
Copyright holder: Miu et al.
License: This is an open access article distributed under the terms of the Creative Commons Attribution License, which permits unrestricted use, distribution, reproduction and adaptation in any medium and for any purpose provided that it is properly attributed. For attribution, the original author(s), title, publication source (PeerJ) and either DOI or URL of the article must be cited.
License URL: https://creativecommons.org/licenses/by/4.0/

Keywords: Protected areas, Spatial prioritization, Biogeographic regions, Designation of strictly protected areas, Systematic conservation planning, Natura 2000, Romania

Funding: Executive Agency for Higher Education, Research, Development and Innovation Funding PN-III-P4-ID-PCE-2016-0483 This work was supported by a grant of Executive Agency for Higher Education, Research, Development and Innovation Funding (PN-III-P4-ID-PCE-2016-0483). The funders had no role in study design, data collection and analysis, decision to publish, or preparation of the manuscript.

==============================
Background

The European Union strives to increase protected areas of the EU terrestrial surface to 30% by year 2030, of which one third should be strictly protected. Designation of the Natura 2000 network, the backbone of nature protection in the EU, was mostly an expert-opinion process with little systematic conservation planning. The designation of the Natura 2000 network in Romania followed the same non-systematic approach, resulting in a suboptimal representation of invertebrates and plants. To help identify areas with very high biodiversity without repeating past planning missteps, we present a reproducible example of spatial prioritization using Romania’s current terrestrial Natura 2000 network and coarse-scale terrestrial species occurrence.

Methods

We used 371 terrestrial Natura 2000 Sites of Community Importance (Natura 2000 SCI), designated to protect 164 terrestrial species listed under Annex II of Habitats Directive in Romania in our spatial prioritization analyses (marine Natura 2000 sites and species were excluded). Species occurrences in terrestrial Natura 2000 sites were aggregated at a Universal Traverse Mercator spatial resolution of 1 km2. To identify priority terrestrial Natura 2000 sites for species conservation, and to explore if the Romanian Natura 2000 network sufficiently represents species included in Annex II of Habitats Directive, we used Zonation v4, a decision support software tool for spatial conservation planning. We carried out the analyses nationwide (all Natura 2000 sites) as well as separately for each biogeographic region (i.e., Alpine, Continental, Pannonian, Steppic and Black Sea).

Results

The results of spatial prioritization of terrestrial Natura 2000 vary greatly by planning scenario. The performance of national-level planning of top priorities is minimal. On average, when 33% of the landscape of Natura 2000 sites is protected, only 20% of the distribution of species listed in Annex II of Habitats Directive are protected. As a consequence, the representation of species by priority terrestrial Natura 2000 sites is lessened when compared to the initial set of species. When planning by taxonomic group, the top-priority areas include only 10% of invertebrate distribution in Natura 2000. When selecting top-priority areas by biogeographical region, there are significantly fewer gap species than in the national level and by taxa scenarios; thusly, the scenario outperforms the national-level prioritization. The designation of strictly protected areas as required by the EU Biodiversity Strategy for 2030 should be followed by setting clear objectives, including a good representation of species and habitats at the biogeographical region level.

Introduction

Protected areas, a critical tool for nature conservation strategies, are intended to ensure the long-term persistence and viability of biodiversity. These areas should support many rare, threatened, or endemic taxa, particularly those with low mobility and high sensitivity to environmental alterations (Geldmann et al., 2013; Gray et al., 2016; Possingham et al., 2006; Rodrigues et al., 2004). When planning protected areas, states around the world are guided by supranational policies such as Convention on Biological Diversity and EU Biodiversity Strategy for 2030, which issue ambitious objectives to increase the extent of protected areas. For example, Convention on Biological Diversity (CDB) Aichi Target on Protected Areas calls for the protection of 17% of the world’s terrestrial and inland water areas in key regions for biodiversity and ecosystem services (UNEP, 2011), while the EU Member States seek to increase by 2030 the Natura 2000 network to 30% of which one third should be under strict protection as areas of very high biodiversity and climate value (European Commission, 2020).

A promising tool to help build an ecologically-sound network of protected areas meeting the CDB or EU targets is systematic conservation planning (Margules & Pressey, 2000). Systematic conservation planning maximizes conservation benefits while minimizing impacts on other resources, such as the availability of productive land. Spatial conservation prioritization, as a part of systematic conservation planning, customarily relies on the complementarity concept (i.e., selection of complementary areas to avoid duplication of conservation effort) and is considered an efficient instrument for identifying spatial priorities and achieving conservation goals (Margules & Pressey, 2000; Pressey et al., 2007).

One of the most extensive networks of conservation areas in the world is the Natura 2000 network, which has been created to operationalize EU Birds (Directive 2009/147/EC, 2020) and Habitats Directives (Council Directive 92/43/EEC, 2020). To date, Natura 2000 encompasses 18% of EU terrestrial area, thus meeting the CDB Aichi Target on Protected Areas (UNEP, 2011). The effectiveness and representativity of Natura 2000 were evaluated for different taxonomic groups and geographic areas, and the conclusions tended to highlight suboptimal planning (D’Amen et al., 2013; Dimitrakopoulos, Memtsas & Troumbis, 2004; Kukkala et al., 2016; Lisón, Palazón & Calvo, 2013; Müller, Schneider & Jantke, 2018; Müller, Schneider & Jantke, 2020; Votsi, Zomeni & Pantis, 2016). The suboptimal planning of Natura 2000 at the EU and at Member States levels originates from an uncoordinated process (Apostolopoulou & Pantis, 2009; Iojă et al., 2010; Lisón et al., 2017; Orlikowska et al., 2016), which was partially resolved by selecting new sites after expert-opinion evaluations during the Natura 2000 biogeographical seminars (Kenig-Witkowska, 2017; Manolache et al., 2017). Furthermore, the efficacy of the Natura 2000 network was extensively re-evaluated from other perspectives, for example, for understanding the effect of climate change on representativity (Araújo et al., 2011; Popescu et al., 2013) and for coordinating conservation investments (Hermoso et al., 2017; Nita et al., 2016).

The designation of the Natura 2000 network in Romania followed the same non-systematic approach. The process started in 2007 with the designation of 273 Sites of Community Importance covering Habitats Directive and 108 Special Protection Areas under Birds Directive. This process continues in the present; nowadays, there are 606 designated Natura 2000 sites (Sites of Community Importance and Special Protection Areas) that encompass 23% of the total country’s land area (54,214 km2) (Directorate-General for the Environment, 2020; Manolache et al., 2017). Of these, 426 are terrestrial Natura 2000 Sites of Community Importance, covering 40,310 km2 (17% of Romania’s terrestrial surface) (Directorate-General for the Environment, 2020; EIONET, 2020). During the first two designation stages, the process was highly biased towards overlapping existing national protected areas (Iojă et al., 2010; Manolache et al., 2017), and thus, even if the CBD 17% target is met, the effectiveness of Natura 2000 in representing habitats and species is questionable. For example, Iojă et al. (2010) confirmed that overlapping existing national protected areas resulted in a suboptimal representation of plants and invertebrates; Miu et al. (2018) highlighted underrepresentation of agricultural landscape in Dobrogea, while Mânzu et al. (2013) and Popescu et al. (2013) concluded that the Natura 2000 network would not protect plants, reptiles, or amphibians if species ranges shift under climate change scenarios.

With the latest extensions, the Romanian Natura 2000 network encompasses all species and habitats listed in Habitats and Birds Directives (Directorate-General for the Environment, 2020; Manolache et al., 2017); however, the new EU Biodiversity Strategy for 2030 requires an increase from 23% to 30% of the total terrestrial country’s area of which one third should be under strict protection as areas of very high biodiversity and climate value (European Commission, 2020). To help identify areas with very high biodiversity and to provide an example of systematic planning of a protected area network, we present a reproducible spatial prioritization case study using Romania’s current terrestrial Natura 2000 network and coarse-scale terrestrial species occurrence (marine Natura 2000 sites and species were excluded). The objectives of this study are (1) to identify candidate sites for designation as areas of very high biodiversity within the Romanian terrestrial Natura 2000 network in national, taxa-specific and biogeographical levels spatial prioritization scenarios and (2) to investigate the extent to which the areas of very high biodiversity within terrestrial Natura 2000 network cover the species listed in Annex II of Habitats Directive in national, taxa-specific and biogeographical levels spatial prioritization scenarios. The European Union assesses the effectiveness of Natura 2000 network in protecting species and habitats listed in Birds and Habitats Directives at the Member State level but also at the biogeographic level (Evans, 2012); thus, we performed the analyses at both administrative levels.

Methods

Natura 2000 sites and species

The dataset used in our planning analysis included 371 terrestrial Natura 2000 Sites of Community Importance (Natura 2000 SCI), designated to protect 164 terrestrial species listed under Annex II of Habitats Directive in Romania. The initial database included 426 Natura 2000 SCI and 166 species (EIONET, 2020), from which we excluded seven sites with a small area (<1 km2, the spatial resolution of our data), 48 terrestrial sites designed only for habitat protection, and two marine species (the common bottlenose dolphin - Tursiops truncatus, and the harbor porpoise - Phocoena phocoena) (Data S1). We used only terrestrial sites and species to match the EU commitment to designate by 2030 as strictly protected one-third of protected areas separately on land and at sea (European Commission, 2020). Of the 164 species protected by this terrestrial Natura 2000 network, 26 are mammals, six are reptiles, six are amphibians, 26 are fish, 54 are invertebrates, and 46 are plants. The number of species protected within a Natura 2000 site varies between one (46 sites protect only one species) and 62 (Iron Gates). The terrestrial sites with the largest number of protected species (>40 species) are Iron Gates, Domogled—Valea Cernei, Calimani—Gurghiu, Danube Delta, Cheile Nerei—Beusnita, Fagaras Mountains, and Tur River. The largest Natura 2000 sites in terms of surface area (>1,200 km2) are Danube Delta, Fagaras Mountains, Frumoasa, Calimani—Gurghiu, and Iron Gates (Rozylowicz et al., 2019).

To map species occurrences in terrestrial Natura 2000, we used site-level occurrence data included in the Natura 2000 Standard Data Forms (EIONET, 2020). Site-level occurrence records were aggregated at a Universal Traverse Mercator spatial resolution of 1 km2 (UTM 1 × 1 km). Thus, if a species was included in the Standard Data Form (recorded for the respective Natura 2000 site), each cell of that site was considered as having that species present. We followed this approach due to the absence of finer scale species distribution data in Romania for Natura 2000 taxa, which makes species distribution modelling impractical for all Natura 2000 taxa. While the coarse spatial resolution likely overestimates the distribution of several range-restricted taxa, data at protected area level, rather than within protected areas, is currently used for official biogeographical assessments of conservation status of species and habitats in Romania under Article 17 of the Habitats Directive (EIONET, 2020). The likely outcome of overestimating the distribution of some species for this prioritization study, which focuses on species-rich areas (Additive Benefit Function algorithm, see section Priority ranking of terrestrial Natura 2000 sites), is that some protected areas may emerge as top priorities despite the fact that some species are occurring only within a relatively small area within those respective protected areas. Thus, the prioritization results should be interpreted as a Natura 2000 site with a certain proportion to be designated as strictly protected areas and not as the exact position of top priority grid cells. As such, this approach closely matches the approach to conservation planning in Romania, which uses species lists to establish protected areas of various sizes and acknowledges that species may only occur in discrete units within a given protected area (EIONET, 2020).

The Natura 2000 sites and species considered here were further sorted by biogeographic region. Because Romania lies at the geographic center of Europe (Rey et al., 2007), Natura 2000 network overlaps five terrestrial biogeographical regions, out of the nine regions recognized by the European Union, i.e., Alpine, Continental, Pannonian, Steppic and Black Sea (Rozylowicz et al., 2019). Due to its small size, the terrestrial part of Black Sea region was merged with the Steppic biogeographic region (Steppic and Black Sea region in our analysis). Of the 164 terrestrial species, 110 are found in the Alpine Biogeographic Region, 143 in the Continental Biogeographic Region, 76 in the Pannonian Biogeographic Region, and 78 in the Steppic and Black Sea Biogeographic Region. Several species were found in 2 or 3 biogeographic regions due to their wide geographic range (e.g., Bombina bombina, Bombina variegata, Emys orbicularis) or because they inhabit a greater range of habitats (e.g., Lutra lutra) (Data S1).

Priority ranking of terrestrial Natura 2000 sites

To identify priority terrestrial Natura 2000 sites for species conservation and to explore these areas adequately representing species included in Annex II of Habitats Directive, we used Zonation v4, a decision-support software tool for spatial conservation planning with Natura 2000 sites as planning units (Lehtomäki & Moilanen, 2013; Moilanen, 2007). We analyzed three priority ranking scenarios: (1) nationwide, (2) nationwide for several taxonomic groups separately (amphibians, reptiles, mammals, fish, invertebrates and plants), and (3) separately for each biogeographic region across all taxonomic groups (Alpine, Continental, Pannonian, Steppic and the Black Sea biogeographic regions) (Fig. 1). Zonation produces a priority ranking by iteratively removing planning units (Natura 2000 sites in our case) with the lowest total marginal loss of conservation value while accounting for total and remaining distributions of protected species (Moilanen et al., 2014). Priority ranking starts from the full Natura 2000 network, and the planning units are iteratively removed until there are none remaining. Least valuable sites (e.g., low species richness) are removed first, while the valuable sites (e.g., high species richness) are kept until the end (Di Minin et al., 2014).

Figure 1 Flowchart illustrating the spatial prioritization process (national, taxa-specific and biogeographical levels spatial prioritization scenarios).

Zonation provides four cell removal rules (Core-area Zonation, Additive benefit function, Target-based planning and Generalized benefit function). For our case study, we used additive benefit function with exponent z = 0.25 (the default value for species sensitivity to habitat loss), a cell removal rule with a summation structure (Moilanen et al., 2014; Moilanen, 2007; Moilanen et al., 2005), which gives higher priority to planning units with a higher number of species present and tended to remove biodiversity-poor cells even if they include rare species. Thus, planning with additive benefit function may result in a selection of top priority areas that have higher performance on average, but retains a lower minimum proportion of original distributions for rare species (Arponen et al., 2005; Moilanen et al., 2014). The additive benefit function fits well to our prioritization objective—identification of high biodiversity value protected areas within the Romanian terrestrial Natura 2000 network.

The outputs of the priority analyses provide the ranking of grid cells within Natura 2000 sites according to their contribution in covering protected species (0 = cells with the lowest priority; 1 = cells with the highest priority). The ranking scores exhibit a uniform distribution; thus, the top spatial conservation priorities (e.g., top 33% of the Natura 2000 network) have a Zonation ranking of ≥0.67. The ranking maps are paired with the performance curves that describe the extent to which each species is retained in any given high-priority or low-priority fraction of the Natura 2000 network (Moilanen et al., 2005; Moilanen, Leathwick & Quinn, 2011). Because we used Natura 2000 as a planning unit layer, our analysis can be used to infer how much of a Natura 2000 site should be designated as strictly protected in order to reach one third objective at network level.

Results

Prioritization of Natura 2000 sites at the national level

Terrestrial Natura 2000 sites represented as 1 × 1 km grid cells cover 48954 cells (20.24% of Romanian territory), of which 4920 overlaps less than 5% with the respective protected area. The Natura 2000 sites with high priority grid cells extend over the Carpathians and Transylvania (Fig. 2). Outside of Carpathians (Eastern and Southern Romania), the top-priority Natura 2000 sites were principally located along river corridors. The sites with the largest number of grid cells labeled as priority for conservation (>400 km2) are Sighisoara Tarnava Mare, Muntii Ciucului, Trascau, Valea Izei si Dealul Solovan, Muntele Ses, Retezat, Podisul Lipovei Poiana Rusca, Dealurile Tarnavei Mici Biches, Semenic Cheile Carasului. A notable exception is the Natura 2000 site overlapping the lower course of Ialomita river in Eastern Romania (Fig. 2, Data S2).

Figure 2 High priority sites for designation as areas under strict protection (national level prioritization scenario).

Grid cells within the Romanian terrestrial Natura 2000 network have been graded according to their priority, with the highest-priority sites (top 33%) shown in red. Biogeographic regions are numbered as follows: I (Alpine), II (Continental), III (Pannonian), IV (Steppic and terrestrial Black Sea).

Top-priority sites in national level scenario cover 37% of Natura 2000 protected grid cells in Alpine biogeographic region, 28% of protected grid cells in Continental biogeographic region, 22% in Pannonian biogeographic region, and 12% of Natura 2000 protected grid cells in the Steppic and the Black Sea biogeographic region (Fig. 2, Data S2).

The performance of national-level planning of top priorities is minimal. On average, when 33% of landscape of Natura 2000 sites is protected, only 20% of distribution of species listed in Annex II of Habitats Directive are protected (Fig. S1). As a consequence, the representation of species by priority terrestrial Natura 2000 sites is lessened when compared to the initial set of species, with 20 species (12%) not covered by the top 33% of protected grid cells. The missed species include plants (12 species), invertebrates (4 species), fish (2 species) and mammals (2 species) (Table 1, Data S3).

Table 1 Representation of protected species by Natura 2000 sites with grid cells selected as high priority in national level prioritization scenario (gap and most represented species).

Taxonomic group	Species	Number of top 33% Natura 2000 sites covering the species	
Plants	Centaurea jankae, Potentilla emilii-popii, Centaurea pontica, Dracocephalum austriacum, Ferula sadleriana, Gladiolus palustris, Stipa danubialis, Thlaspi jankae, Tulipa hungarica, Paeonia officinalis subsp. banatica, Colchicum arenarium, Saxifraga hirculus	0	
	Ligularia sibirica	17	
Invertebrates	Graphoderus bilineatus, Stenobothrus eurasius, Isophya harzi, Vertigo moulinsiana	0	
Lucanus cervus	41	
Fish	Cobitis elongata, Rutilus pigus	0	
Barbus meridionalis	62	
Amphibians	–	0	
Bombina variegata	108	
Reptiles	–	0	
Emys orbicularis	24	
Mammals	Mustela lutreola, Rhinolophus mehelyi	0	
Lutra lutra	110	

Prioritization of Natura 2000 sites at national level by taxonomic group

When priority ranking maps are organized by taxonomic group, the top-priority sites are dissimilar to the results of national level prioritization (Fig. 3). For amphibians, the ranking map (Fig. 3B) indicates that sites with high priority grid cells for conservation are spatially grouped in the western and central parts of Romania, while for reptiles, the sites were clustered in the southwestern and southeastern part of Romania (Fig. 3B). For mammals and plants, most of the high-priority sites are spatially grouped in the Carpathians and Dobrogea areas, regions with large protected areas and high species richness. High priority Natura 2000 sites for invertebrates and fish are dispersed within the country (Figs. 3C–3F).

Figure 3 High priority sites for designation as areas under strict protection (taxonomic level prioritization scenario).

Grid cells within the Romanian terrestrial Natura 2000 network have been graded according to their priority, with the highest-priority sites (top 33%) shown in red. Biogeographic regions are numbered as follows: I (Alpine), II (Continental), III (Pannonian), IV (Steppic and terrestrial Black Sea). (A) Amphibians. (B) Reptiles. (C) Invertebrates. (D) Fish. (E) Mammals. (F) Plants.

The performance of top-priority Natura 2000 sites in representing species distribution varies by taxonomic group (Fig. S2). For invertebrates, the prioritization ranking indicates that if the top 33% of the landscape included in Natura 2000 sites is strictly protected, on average only 20% of the invertebrate distributions in Natura 2000 are also protected; this is followed by the amphibian group, with over 50% of the amphibian distribution in Natura 2000 protected. For reptiles, Natura 2000 performs better, with over 90% of the distribution of reptiles in strictly protected areas. For mammals, fish, and plants, the Romanian Natura 2000 network performs very well, with more than 75% of distribution of the respective species strictly protected when the identified top priority 33% of Natura 2000 area is protected.

Prioritization of Natura 2000 sites at biogeographical level

Within the Alpine biogeographic region, the high-priority areas are in Western Carpathians (Natura 2000 sites overlapping Trascau and Apuseni Mountains), Southern Carpathians (Natura 2000 sites overlapping Retezat, Domogled, Cerna, Cozia Mountains), and Eastern Carpathians (Natura 2000 sites overlapping Maramures and Gurghiu Mountains) (Fig. 4A, Data S2). Within the Continental biogeographic region, the main high-priority areas overlap Natura 2000 sites in Iron Gates, Semenic, Cheile Nerei (SW Romania), Sighisoara Tarnava Mare (center of Romania), Dealurile Clujului de Est (center of Romania), Tur River (northwest of Romania), and Ciuperceni - Desa (south) (Fig. 4B, Data S2). In the Pannonian biogeographic region, the main high-priority area is in the Lower Mures Floodplain and Carei Plain (western part of Romania) (Fig. 4C, Data S2). In the Steppic and terrestrial Black Sea biogeographic regions, the high-priority areas are North-Dobrogea Tableland, Macin Mountains, Canaralele Dunarii, and Padurea si Valea Canaraua Fetii –Iortmac and Lower Siret Floodplain (Fig. 4D, Data S2).

Figure 4 High priority sites for designation as areas under strict protection (biogeographical level prioritization scenario).

Grid cells within the Romanian terrestrial Natura 2000 network have been graded according to their priority, with the highest-priority sites (top 33%) shown in red. (A) Alpine biogeographic region. (B) Continental biogeographic region. (C) Pannonian biogeographic region. (D) Steppic and terrestrial Black Sea biogeographic region.

The performance of Natura 2000 by biogeographical region is similar for Alpine, Steppic & terrestrial Black Sea and Pannonian. In these regions, when the top 33% of the respective region is strictly protected, between 55% and 65% of the distribution of covered species is strictly protected. The Continental biogeographic region performs better when considering that 33% of landscape of Natura 2000 sites are strictly protected; nearly 75% of the distribution of species listed in Annex II of HD are strictly protected (Fig. S3).

The biogeographic level planning scenario produced fewer gap species (i.e., not covered within the strictly protected area network) than the national level scenario when selecting top priorities (Fig. 5, Data S3). Out of 118 species included in Alpine biogeographic region, 9 are gap species if planning is done at the national level and 3 if planning is done at the biogeographical level. Of 149 species in Continental biogeographic region, 37 are gap species when prioritization is done at the national level and 6 when it is done at the biogeographical level. Furthermore, out of 75 species represented in Pannonian biogeographic region, 27 are gap species in national-level scenario and only 3 in biogeographical level scenario. Also, the number of gap species is reduced when planning is done at the Steppic and terrestrial Black Sea biogeographic region level, with no gap species out of 38 in national-level scenarios and only 12 species in biogeographical level scenarios.

Figure 5 Species representation in the top 33% planning units by prioritization scenarios.

(A) Amphibians, (B) Reptiles, (C) Invertebrates, (D) Fish, (E) Mammals, (F) Plants. Red = gap species (never included in Natura 2000 sites with high priority grid cells in the respective scenario), Blue = covered species (included in Natura 2000 sites with high priority grid cells in the respective scenario). RO = national level scenario. ALP RO = Alpine EBR in national level scenario. ALP BR = Alpine EBR in biogeographical level scenario; CON RO = Continental EBR in national level scenario. CON BR = Continental EBR in biogeographical level scenario; PAN RO = Pannonian EBR in national level scenario. PAN BR = Pannonian EBR in biogeographical level scenario; STE RO = Steppic and terrestrial Black Sea EBR in national level scenario. STE BR = Steppic and terrestrial Black Sea EBR in biogeographical level scenario.

Discussion

The EU Biodiversity Strategy for 2030 acknowledges that the current network of protected areas, including those under strict protection, is not sufficiently large to safeguard Europe’s biodiversity in the face of multiple stressors. To overcome this issue, the European Commission set ambitious conservation objectives for Member States, such as the enlargement of protected areas to at least 30% of terrestrial national territory in Europe, of which one-third should be strictly protected (European Commission, 2020). To support policymakers in establishing criteria and guidance for meeting the objective of one-third of protected areas under strict protection, we tested three spatial conservation prioritization scenarios using the Romanian terrestrial Natura 2000 network as a case study. Our analyses suggest that selecting strictly protected areas at the European Biogeographical Region level performs better than nationwide or taxa-specific planning scenarios in terms of species representation and spatial evenness of selected sites.

The EU Biodiversity Strategy for 2030 outlines key principles designation of areas of very high biodiversity value, such as including carbon-rich ecosystems (old-growth forests, peatlands) or outermost regions. The strategy also stipulates potential planning scenarios, e.g., at EU biogeographical regions, national level (European Commission, 2020). Because the backbone of strictly protected areas will be within the Natura 2000 network, which already covers over 17% of EU land, these areas with high biodiversity value should also ensure the long-term survival of species and habitats listed in Birds and Habitats Directives (Evans, 2012), therefore contributing to the implementation of the two Directives by the EU Member States.

In the case of Romanian terrestrial protected areas, when spatial conservation prioritization is done at the national level, the top 33% protected grid cells cover, on average, less than 18% of Habitats Directive-listed species occurrences within the existing Natura 2000 (see Fig. 2, Data S2). The limited coverage of most species indicates that prioritization at a national level is insufficient to ensure that the favorable conservation status is maintained for most species listed in Habitats Directive. Most species that would not be represented in strictly protected areas are plants; 46 plant species will be strictly protected in areas of less than 10 km2, which may be sufficient only for some range-restricted species since efficient management requires specific measures such as fencing or manual mowing (Heywood, 2019). National-level prioritization will also lead to a lack of representation of endangered mammals, such as the European mink Mustela lutreola and the marbled polecat Vormela peregusna. Under the national level prioritization, most of Natura 2000 sites are located within the Continental biogeographic region (51% of priority areas and 48% of the region). The Alpine biogeographic region, which harbors most of the remaining old-growth forests in the Carpathian Mountains (Veen et al., 2010), had a relatively low contribution to high-value priority areas (27% of Alpine region) (see Data S1). The Continental region, which in this scenario would constitute the backbone of strictly protected areas, include many common species when compared to other biogeographic regions (Gruber et al., 2012; Rozylowicz et al., 2019); thus, rare species inhabiting Alpine, Steppic and terrestrial Black Sea regions would not be represented in strictly protected areas without a significant expansion of protected areas network. This finding corroborates previous work that found that more than 50% of sites from Alpine, Steppic, and terrestrial Black Sea regions are important for the cohesion of the Natura 2000 network at the national level (Rozylowicz et al., 2019).

The limited contribution of the national-level prioritization scenario may be due to the prioritization algorithm selected for this analysis (Additive Benefit Function), which favors Natura 2000 sites with high species richness (Di Minin et al., 2014). Using other removal rules, such as Core-Area Zonation, which strives to provide the best representation for each individual species, would result in better representation of range-restricted species; as a result, Natura 2000 sites with a high number of endemic species would be retained as areas of high biodiversity value (Kukkala et al., 2016); however, forests and other important carbon-rich ecosystems would be missed, thus limiting more their contribution to achieve the EU Biodiversity Strategy for 2030 of one-third under strict protection objective.

Spatial conservation prioritization relies on the quality of species distribution data (Wiersma & Sleep, 2016). Studies typically opt to limit planning exercises to the best available species data set (e.g., Kukkala et al., 2016; D’Amen et al., 2013); however, drawing conclusions on data-rich taxa likely limits the application of systematic conservation planning at a continental level that consider species across all taxonomic groups (see Jung et al., 2020 for a comprehensive global analysis). Our analysis, while coarse, does explore several taxonomic groups (amphibians, reptiles, fish, mammals, invertebrates, plants), thus providing a national-level perspective on protecting many levels of biodiversity. For example, we found a limited value of applying spatial conservation prioritization algorithms at national level by taxonomic group (see Fig. 3, Fig. S2, Data S2). In our taxa specific scenarios, the top 33% priority areas overlap 295 Natura 2000 sites, of which 222 sites include priority areas for invertebrates, 92 sites for amphibians, 73 sites for plants, 63 sites for fish species, 58 sites for reptiles, and 42 sites for mammals. Achieving the EU Biodiversity Strategy targets will result in overshooting the one-third strictly protected target, and will require significant land availability and funding to implement, neither being a feasible and efficient prerequisite to conservation in Romania and the EU (Hermoso et al., 2019). Despite their limited value for the EU biodiversity targets, the taxonomic group-based scenarios can be used to identify key areas for a specific taxon and could be used to complement the more realistic, biogeographic regional-based scenarios. This type of prioritization can also be used to understand the data gaps across taxonomic groups. For example, in our case, the top 33% planning units for 53 invertebrate species include 222 Natura 2000 sites out of 371, while for 46 plant species, the prioritization algorithm will select planning units for 73 Natura 2000 sites. The large number of sites selected in the top 33% for invertebrates is a direct result of insufficient monitoring efforts for these species, and a lack of taxonomists (Brodie et al., 2019; Cardoso et al., 2011; D’Amen et al., 2013). Data gaps likely resulted in a lower than expected species per site, thus affecting the outcomes of our prioritization exercise.

The biogeographical region level prioritization resulted in a balanced distribution of top-priority planning units across the country. This is an expected result, as prioritization using biologically-significant administrative borders will reduce the lower coverage of areas with many range-restricted species (Kukkala et al., 2016). The biogeographic region level planning scenarios also resulted in a smaller number of sites with planning units in top 33%, with 104 Natura 2000 sites when planning region by region (19 sites belong to more than a region) compared to 222 sites under the national level scenario (see Fig. 4, Data S2). Most sites with top-priority grid cells occur in the Continental region (56 sites) and the Alpine region (38 sites), followed by the Steppic and terrestrial Black Sea (23 sites), and the Pannonian regions (8 sites). Biogeographic-focused planning scenarios also performs better in terms of species representation, with only 12 species not covered by top-priority planning units (see Data S3, Fig. 5). Only one species is missed by all biogeographic regions - European bison, Bison bonasus) - which would require only one new Natura 2000 site for complete representation (e.g., Tarcu Mountains in SW Romania is on of them, and there are ongoing efforts to reintroduce bison in the Southern Carpathians, Fagaras Mountains).

Our prioritization is constrained by the limited availability of occurrence data for most of the Annex II Habitats Directive species. With few exceptions, such as reptiles, amphibians, mammals in the Dobrogea region, large carnivores (Bîrsan et al., 2017; Cogaˇlniceanu et al., 2013a; Cogaˇlniceanu et al., 2013b; Cristescu et al., 2019; Miu et al., 2018) species distribution data are available as extent of occurrence, rather than specific locations or modeled species distributions (EIONET, 2020). Also, other sources extensively used in prioritization research, such as GBIF (e.g., Guo et al., 2020), include low numbers of occurrence data for Romania. To overcome this shortcoming we used Natura 2000 Standard Data Form, the technical documentation of a Natura 2000 which includes species for which it was designated (Lisón et al., 2017), and the spatial resolution of data reported by the Romanian authorities for biogeographical assessments of the conservation status of species and habitats under Article 17 of the Habitats Directive (EIONET, 2020). While this is less than ideal for systematic conservation planning, it showcases the real-world decision-making process in Eastern European conservation. This is why we support the existing calls for obtaining robust species distribution data prior to establishing and planning strictly protected areas, especially for overlooked species such as invertebrates (Cardoso et al., 2011).

Conclusions

The EU Biodiversity Strategy for 2030 requires an expansion of protected areas network in Europe, of which one third should be under strict protection as areas of very high biodiversity and climate value. The strategy outlines key principles for designation of strictly protected areas but without providing clear guidelines. To support policymakers in establishing criteria and guidance for meeting the target of one-third of protected areas under strict protection, we provide here a reproducible spatial prioritization case study using Romania’s current terrestrial Natura 2000 network and coarse-scale terrestrial species occurrence. Our results indicate that designation of strictly protected areas using a systematic conservation planning approach at biogeographic region-level would result not only in a good representation of all species protected by EU legislation in a country but also in spatial evenness of selected sites. The species-specific approach used in our example may be easily expanded to include other dimensions of biodiversity, such as carbon-rich areas and old-growth forests, ecological corridors, etc. However, because the results are dependent not only on setting clear targets but also on data quality, we urge policymakers to invest in producing high-quality biodiversity data before proceeding to the designation of new areas of strict protection.

Supplemental Information

Supplemental Information 1 Terrestrial Natura 2000 sites and species included in the study

Click here for additional data file.

Supplemental Information 2 Terrestrial Natura 2000 sites of Romania with grid cells in top 33% by priority ranking

Prioritization scenarious: top 33% all ROU = national-level scenario; top 33% ALP biog = Alpine EBR scenario; top 33% CON biog = Continental EBR scenario; top 33% PAN biog = Pannonian EBR scenario; top 33% STE&BLK biog = Steppic and terrestrial Black Sea scenario.

Click here for additional data file.

Supplemental Information 3 The number of Romania’s Natura 2000 sites protecting species listed under the Habitats Directive (Annex II species)

All sites and top 33% in national and biogeographical levels scenarios.

Click here for additional data file.

Supplemental Information 4 Coverage of species distribution within Romanian terrestrial Natura 2000 network at various proportions of strictly protected landscape

National level prioritization scenario. The curve represent mean coverage achieved across all species.

Click here for additional data file.

Supplemental Information 5 Coverage of taxonomic group distribution within Romanian terrestrial Natura 2000 network at various proportions of strictly protected landscape

Taxonomic group level prioritization scenario. The curves represent mean coverage achieved across all species of the respective taxonomic group.

Click here for additional data file.

Supplemental Information 6 Coverage of species distribution within biogeographical regions overlapping Romanian terrestrial Natura 2000 network at various proportions of strictly protected landscape

Biogeographical level prioritization scenario. The curves represent mean coverage achieved across all species.

Click here for additional data file.

We thank the two reviewers for comments and suggestions and Edward F. Rozylowicz for proofreading and suggestions, which helped us to improve the quality of the manuscript.

Additional Information and Declarations

Competing Interests

Author Contributions

Data Availability

The authors declare there are no competing interests.

Iulia V. Miu and Laurentiu Rozylowicz conceived and designed the experiments, performed the experiments, analyzed the data, prepared figures and/or tables, authored or reviewed drafts of the paper, and approved the final draft.

Viorel D. Popescu and Paulina Anastasiu conceived and designed the experiments, authored or reviewed drafts of the paper, and approved the final draft.

The following information was supplied regarding data availability:

The data underpinning the prioritization analyses are available at Zenodo: Miu, Iulia V, Rozylowicz, Laurentiu, Popescu, Dan V, & Anastasiu, Paulina. (2020). Data from: Identification of top-priority areas to achieve EU Biodiversity Strategy for 2030 key commitments (Version 2) [Data set]. Zenodo. http://doi.org/10.5281/zenodo.3999171.

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
