# Peer review of "Identification of areas of very high biodiversity value to achieve the EU Biodiversity Strategy for 2030 key commitments"

_PeerJ, doi:10.7717/peerj.10067_

## Round 0.1 · original submission · Major Revisions

Please address the comments made by the reviewers. I look forward to receiving the revised manuscript.

Reviewer 1 ·

Basic reporting

1. There are several sections where the writing could be improved for clarity. I have provided specific examples below.
Line 20: remove “the” in “the protected areas”
Line 21: rewrite to area of high biodiversity value
Line 22: add “the” after “of”
Line 30: Starting the methods section of the abstract with “the planning exercise” is vague and confusing. It would help if more specific language was used such as “We used 315 terrestrial… in our spatial prioritization analysis”.
Line 32: Is there a difference between terrestrial Natura 2000 and Natura 2000? Please clarify
Line 33: should be km2
Lines 40-42: These sentences need more context to be understood by a reader that has not read your whole paper already.
The authors overuse the phrase “when considering” and in most cases it can be dropped from the sentence to improve readability (example on line 40-41, line 226, line 235, etc.)
Line 54-55: rewrite to “They should support as many rare, threatened or endemic taxa, particularly those with low mobility and high sensitivity to environmental alterations as possible”
Line 63: “by 2030 to 30%” is awkward. I suggest rewriting so the year and percentage are not beside each other.
Line 78: I suggest cutting “and except for an agreement on better than random planning performance” as it does not add much to the sentence and makes it harder to read.
Line 123: which dataset? Please be specific
Line 126: I think you mean < 1km2
Line 138: change records to record
Line 147-147: rewrite to: “the Natura 2000 network overlaps”
Line 165: Please merge this paragraph with the previous
Line 165: what is meant by grid cells?
Line 166: add “the” after “for”
Line 167: what do you mean by “remaining characteristics of the analysis?
Line 171: Please explain what is meant by “between the characteristics”
Line 173-174: please explain what the additive benefit function is. Non-expert readers may not be familiar with this function.
Line 174: please remove “representing the exponent”
Line 176: please explain this line for a non-expert reader.
Line 178: consider changing “covering” to “conserving”
Line 179-180: The sentence beginning with “Zonation” is repetitive and can be removed.
Line 186: please remove “falling”
Line 193-194: I am unclear what is meant by this sentence. Would you please clarify and elaborate on this result further. I do not understand the difference between the first set of planning units (48954) and the second set of planning units (4920)
Line 305: what is meant by “poor coverage of invertebrates”? and “Invertebrates are overlooked”. Please elaborate.
Line 327: do you mean occurrence data not distribution data?
Line 341: I don’t know if scale is the right word to look at here, but perhaps this could be rewritten to it is important to focus on biogeographical context when planning priority areas?
2. The authors provide a good amount of background information and the manuscript is well cited. However, I think the authors should include why including biogeographic regions is important in the introduction as it is part of their research question, yet it is not mentioned in the introduction.
3. The structure of the discussion needs improvement. The authors mainly restate their results within the discussion without adding much additional interpretation.
The first paragraph of the discussion should begin with the research objective then quickly summarize the main findings of the paper. Currently, there is too much emphasis introducing the research question and only one results point is mentioned.
Each paragraph in the discussion is missing an explanation of the relevance or significance of the findings leaving the reader wondering “so what”? What does it mean that invertebrates are driving the taxa specific example? What can be done about the lack of invertebrate surveys? What is the significance of the biogeographical scenario performing the best? Why might this be the case? Have other studies found this as well?
The conclusion section is also very limited in details and needs to be elaborated on (see validity of findings section for more details on the conclusion).
4. Table 1: it is unclear which scenario this data is referring to. Please clarify in the legend. Also, it is hard to follow the table currently. Consider adding horizontal lines or increasing the spaces between rows so that it is easier to read. Does a zero mean that for all those species in the row they were never prioritized? Please clarify.
The figures are well drawn and are easy to interpret.

Experimental design

1. The research question is well defined and is an important question to address. This study addresses an important knowledge gap.
2. I would like to see more information provided about where the species occurrence data came from and how it was collected, and at what year was the data collected. Did the authors consider other sources of species occurrence data such as GBIF or iNaturalist which could increase resolution of the species surveys, which is a major limitation of this study. If not, why did the authors choose not to use other data sources?
3. The methods appear sound and are generally well reported, but may need to be explained in more plain language for non-experts to understand. I provided specific examples in the basic reporting section.

Validity of the findings

1. The findings appear to be valid, but the implications of the findings need to be elaborated upon (see basic reporting section). The conclusion needs to be elaborated upon to be more specific and to provide concrete examples of future work that could be conducted and the relevance of the study. The finding that biogeography needs to be considered and that current efforts looking at the national scale will perform poorly is very interesting and will be extremely important for designating future protected areas and I think that needs to be really highlighted within the conclusion section. Consider the following for expanding your conclusion section: what will the outcome be if non-systematic planning continues in the future? What will the outcome be if a national scale is used? What about the importance of continued species monitoring to improve occurrence data and how might that influence future decisions?

Additional comments

This study is very interesting and topical and the questions posed are important and need to be addressed. The authors did an excellent job making the case for the need to consider biogeography in planning protected areas.

·

Basic reporting

The introduction was structured well and gives sufficient detail to ground this study.
The methods made structural sense as well, but I do have some questions about methodological details (see experimental section) that I think need to be addressed to make sure readers can follow what the authors have done.
The rest of the manuscript is structured well, with appropriate literature cited.

I am not sure if the data shared would be sufficient to replicate the analysis. I generally hope for authors to share their data and analysis scripts, so that others (including reviewers) can properly verify analysis and comment on reproducibility of the analysis underlying a manuscript meaningfully.

Experimental design

I’ve read the methods section three times now and I’m still not sure I understand how the authors have setup their analysis. Some questions that I still have:

Was each 1x1 km pixel of the country used as planning units, or only those that were Natura 2000 sites? I think it was only Natura 2000 sites, but am not sure. Could you clarify that in text?

Starting on line 140 you state: “If a species was included in the Standard Data Form for a given site, it was considered as present in all UTM 1 × 1 km overlapping the respective site.”

Does that mean that if a species was recorded for a Natura 2000 site, each cell of that site was considered as having that species present? To me this is a huge assumption and worrisome. If I think about the fact that some sites are >1200km2, I worry that this could easily be more area than the entire species might occupy. You mention this in the last paragraph of the discussion, but I was hoping you could expand on this in terms of what this would mean for the Zonation analysis as well? I think this would be important to note for reader to fully understand the potential limitations of this data set.

I think making sure readers understand what data was used how is very important to understand both, what the authors did and equally important, what the author’s findings mean.

Validity of the findings

This section is hard for me to comment on, as I did not fully understand the experimental design portion of this review.

I will still comment on two aspects of the work:

1. If the analysis has indeed been conducted at the Natura 2000 site level, I am wondering what the analysis would actually show? If this analysis is restricted to Natura 2000 sites, the analysis can only rank Natura 2000 sites and does not tell us anything about the broader question of whether Natura 2000 sites are effective at capturing high biodiversity areas across Romania. To me, the ranking of existing Natura 2000 sites isn’t that informative. This might very well be my not fully understanding what the authors have set out to do here.

Could you please add text explaining why you think this approach makes sense and what it actually tells us?

If the assumption (and its just my assumption, as I don’t fully understand the methods as currently written) is correct that an occurrence of a species at a site would result in the entire site to be coded as having the species present, I don’t quite understand how the analysis can yield results on high biodiversity areas requiring Zonation analysis. Could you please add some text to help clarify your thinking about this?

2. If the analysis was indeed conducted across the entire country, it would be good to make this clear in methods and also results.

Additional comments

It would be great if you could share the Zonation input files and analysis description/script, so we (reviewers and readers) could replicate your work to help us understand how your analysis was conducted.

Figure 1: Could you make the biogeographic region delineation easier to see, by changing the line color or make them wider? Its hard to make out the different regions at the moment.

Best regards,
Richard Schuster

Research Associate, Carleton University
Email: [email protected]

---

## Round 0.2 · accepted · Accept

Many thanks for clearly addressing the reviewers' comments.

·

Basic reporting

-

Experimental design

-

Validity of the findings

-

Additional comments

Thank you very much for carefully addressing all reviewer comments.

I am happy with how you responded to all comments and don't have any further comments and think the manuscript can be published as is.

Reviewer 3 ·

Basic reporting

English is sufficient.
Line 78-81 is about Natura 2000 effectiveness.
In 2019, work on this topic appeared: Effectiveness of Natura 2000 areas for environmental protection in 21 European countries (open access). You can find interesting information about the state of Natura 2000 areas in Romania. This is just a suggestion.
The literature references are sufficient.
The article structure is good, as well as all figures, tables, and raw data.

Experimental design

Original primary research within Aims and Scope of the journal - the submission is relevant and meaningful.
Methodology is correctly described (especially after comments from previous reviewers).

Validity of the findings

The results are very interesting in the context of the EU Biodiversity Strategy for 2030 key commitments.

Additional comments

I have read the first draft of the manuscript and the comments of previous reviewers. In my opinion, most of the comments were properly taken into account in the text improvement process.